# The Naked Mole-Rat: An Unusual Organism with an Unexpected Latent Potential for Increased Intelligence?

**DOI:** 10.3390/life9030076

**Published:** 2019-09-16

**Authors:** Dirk Schulze-Makuch

**Affiliations:** 1Astrobiology Group, Center for Astronomy and Astrophysics (ZAA), Technical University Berlin, 10623 Berlin, Germany; 2School of the Environment, Washington State University, Pullman, WA 99163, USA

**Keywords:** eusociality, intelligence, adaptation, evolution, extreme environment, animal

## Abstract

Naked mole-rats are eusocial, hairless mammals that are uniquely adapted to their harsh, low-oxygen subsurface habitat. Although their encephalization quotient, a controversial marker of intelligence, is low, they exhibit many features considered tell-tale signs of highly intelligent species on our planet including longevity, plasticity, social cohesion and interaction, rudimentary language, sustainable farming abilities, and maintaining sanitary conditions in their self-built complex housing structures. It is difficult to envision how naked mole-rats would reach even higher levels of intelligence in their natural sensory-challenged habitat, but such an evolutionary path cannot be excluded if they would expand their range onto the earth’s surface.

## 1. The Naked Mole-Rat—An Unusual Eusocial Mammal

Naked mole-rats (Figure 1) are multicellular extremophilic, mouse-sized rodents, which are well-adapted to the harsh, hypoxic conditions commonly found below the ground [1]. They are renowned for their exceptional longevity [2] and resistance to many age-associated chronic diseases [3,4]. For example, in contrast to mice, cancers are very rarely observed in naked mole-rats [5,6,7,8], as well as their pronounced resistance to stressors that are commonly lethal to other small rodents [9]. In their natural habitat in the arid regions of sub-Saharan Africa, they live communally in colonies of up to 300 individuals in an extensive maze of interconnected chambers and burrows, which can be several kilometers long, and may reach depths of 2 m below the surface [10,11].

Naked mole-rats are one of only two known eusocial mammals, the other being a close relative, the Damaraland mole-rat, *Fukomys damarensis*. Like the well-characterized eusocial insects—ants, termites, and bees—reproduction is restricted to a single breeding female and 3–4 male consorts. They form stable long-lasting colonies that are well maintained by communal activities of the subordinates, sacrificing their own reproductive fitness for the greater good of the colony. Although only the queen produces offspring and nurses the pups, all animals within the colony participate in “childcare”, they provide solid foods and inoculate the juvenile microbiome, by producing, on demand, fecal pellets when pups initiate coprophagic begging behaviors. Should the colony be alarmed, the non-breeding adults will carry pups in their mouths away from the source of danger. Conversely, should a pup stray too far from the nest, their older siblings will carry them back to the nest.

The entire colony rests together in deep nests, keeping young pups warm and reducing the need for facultative thermogenesis and minimizing individual heat loss [12]. Even with a reduced resting metabolic rate, respiring together in deep football-sized nests, naked mole-rats rapidly exhaust available oxygen in the poorly ventilated underground atmospheres. As a result, these resilient rodents commonly encounter prolonged periods of oxygen deprivation. Oxygen levels are usually at about 2%–9% in these burrows rather than the 21% we humans enjoy at the surface. It has been shown that they can survive zero percent oxygen for at least 18 min with no apparent ill effects [13]. In addition, experimental studies using electrophysiological ex vivo assessments of brain function, reveal the brain can fully recover after 30 min without access to any oxygen, a feature attributed to a blunted calcium response to low oxygen [14,15]. Under normoxic oxygen conditions, the basal metabolic rate of the naked mole-rat is 75% of that of a mouse [16], however, and they can drop their metabolic rate even further when oxygen levels are extremely low.

Finding food below ground, especially in the sub-Saharan arid regions where food resources are very sparse and patchily distributed, may have been a driving force for the evolution of a eusocial colonial lifestyle where numerous individuals go out in different directions in search of foods which, if found, they bring back to the nest for other conspecifics to share [17]. In keeping with the concept of an extended organism, subordinates communally harness energy resources for use by the colony as a whole. Animals forage in groups of 3 or 4 by tunneling through opaque soil, extending their burrows and in search of underground foods. This process is energetically costly, and given the patchy distribution of foods in these arid areas, foraging success is low and all animals share the rewards of this task [18]. Naked mole-rats are strictly herbivorous, meeting all their nutrient and water needs by the consumption of bulbs, corms, roots and predominantly, the large tubers of the plant *Pyrenacantha kaurbassana.* These tubers may be more than 30 kg in weight, can be found at depths of 25–50 cm and can provide sufficient nutrients to sustain an entire colony [10].

## 2. Can Naked Mole-Rats be Considered Intelligent?

Intelligence is difficult to quantify or define even within the same species. Usually, animals that have to range widely for food and are more selective in regard to their food sources also tend to show higher intelligence [19]. These animals need to invest more into memory to remember where the last meal was than, for example, ruminants such as cows and goats which eat nearly any type of plant. In particular, animals that live in relatively food-poor environments, such as bears or the great apes, that have to range across a wide territory to find their specialist foods, would have an advantage if they could remember where food was this time last year, not just where they found it yesterday [20,21]. Thus, we would expect intelligent animals to be, generally, those species that have wider ranges. This may also be true for the naked mole-rat, needing to not only maneuver through their complex multi-level burrow system, but also needing to recall where their large tubers and other food stashes are located. Thus, we would expect naked mole-rats to exhibit a suite of complex cognitive abilities. Cognitive abilities are usually assumed to be dependent upon the absolute size of the brain and the relative brain size to that of body mass or the encephalization quotient (EQ), with larger brain sizes generally assumed to be indicative of greater intelligence [22]. While this relationship might hold true for primate species, being indicative of more neurons, even within the plethora of data for humans, the correlation between EQ and intelligence remains highly controversial. As a coarse indicator of cranial capacity, it is generally not a good measure of species’ cognitive abilities. For example, while it is generally agreed that whales, dolphins and porpoises are highly intelligent organisms, their measured EQ is relatively low, primarily because they have a lot of fat/blubber for insulation in the cold waters that they inhabit, where heat loss through thermal conductance is high. If we had some way to quantify and account for the amount of fat that contributes to the body weight, their EQ would be substantially higher, more comparable to other species, and in a range to do justice to their intelligence. Similarly, birds, despite numerous markers of intelligence, tend to have small brains for their body size. The underlying reason is because birds need to fly and thus, their brains are structured differently from mammalian brains with more nerve cells and fewer supporting cells per gram of brain. In fact, corvids and some parrots have the same or greater forebrain counts of neurons as primates with much larger brains [23]. Again, if we had some way to quantify and account for the effect of the structural differences, the EQ of birds would be substantially higher, better comparable to other species, and do justice to their intelligence. The naked mole-rat, in keeping with its small body size, has the smallest brain size of the subterranean rodents and has a smaller brain than expected for a rodent of its body size and a substantially lower measured EQ than that of mice. It also has a lower neocortex/brain ratio and less neurons than predicted for its body size [24]. Despite the fact that its social lifestyle demands that naked mole-rats be able to recognize individuals and coordinate various colonial behaviors, eusociality does not appear to drive the evolution of neuron-dense or larger brains. Based upon these criteria, we would not consider naked mole-rats as particularly intelligent animals.

However, because the EQ has been shown to not be a good measure evaluating a species´ intelligence [22], it might be instructive to also discuss various other parameters promoting intelligence, especially given the objective to evaluate whether there is a possible evolutionary path for naked mole-rats to become intelligent in the future and whether a similar organism on an alien world could have developed advanced intelligence. One factor generally believed to promote intelligence is body size. Most of the intelligent animals have larger body sizes compared to the average in their taxonomic group, such as elephants or cephalopods. There are exceptions, though, such as the red fox (*Vulpes vulpes*), well-known for its intelligence and cunningness [25]. Naked mole-rats, the smallest of the more than 50 species of subterranean mammals, lie on the opposite end of the spectrum, a trait linked at least in part, to their harsh habitat and possibly linked also to their extreme longevity for low levels of growth hormone and insulin-like growth factor are commonly associated with increased lifespan [14].

Another commonality of intelligent animals is that they have an active lifestyle. As a consequence of that lifestyle, they have a high degree of sensor processing and motor control, especially in regard to feature extraction, depth perception and distinguishing foreground from background, memory, acceleration and balance, spatial orientation, and coordination of muscle activity [26]. Good examples here are dolphins and humans. Naked mole-rats rank high here because they are highly active and orient themselves very well spatially, albeit within the confined space of self-built, dark, dank burrows.

The ability to process sophisticated sensory input is important for intelligent animals. Examples are primates and parrots that live in complex arboreal habitats and cetaceans with their excellent hearing and echo-locating abilities. Naked mole-rats have an extensive repertoire of distinct vocalizations, which enable them to communicate with each other at close range. Eighteen of these distinctive sounds are within the human hearing range and include vocalizations when food is provided, unique calls associated with mating and fighting, and when alarmed by a predator, a breech in the colony structure or an invasion from a non-related mole-rat [27]. There are also several different distinct calls made by juveniles when outside the nest and when looking for the lactating queen. Lacking external ear pinna, sound localization is poor and since low-frequency sounds and vibrations propagate better underground, naked mole-rats appear to be more sensitive to lower frequency sounds and the click-like sounds generated when their teeth rub together for longer-distance communication. However, anatomical studies have revealed an unexpectedly extremely small middle ear cavity for a 35–40 g rodent, suggesting that long-distance, low-frequency hearing is poor, and that naked mole-rats do not rely heavily on sounds to navigate through their burrow system [28]. Similarly, studies involving conditioned avoidance reveal an extremely restricted low-frequency hearing range and attenuated function of the binaural auditory brain stem neurons [29] suggesting naked mole-rats have poor long-distance hearing.

Visual cues are also of limited use in the dark underground milieu. Given the high metabolic costs associated with a highly functioning visual system, it is not surprising that naked mole-rats have a very small eye with a disorganized retinal arrangement [30], degenerate neural structures associated with vision [31,32] and even, from a very young age, have cataracts as indicated by opaque eye lenses [5]. The optic nerve is also extremely thin, and it appears that while naked mole-rats are capable of detecting light and dark, they are unable to form clear images. Arranged, tactile sensory hairs help the naked mole-rats orientate themselves. Not surprisingly, unlike other mammals, a larger proportion of the neocortex is dedicated to the processing of tactile information rather than vision [33,34]. Lack of visual stimuli and the concomitant attenuated need for a visual cortex may have contributed to their smaller than anticipated brain size without compromising their intelligence.

Intelligent animals such as octopi, parrots, and humans can act on the high sensory input with fine motor control which includes the coordination of multiple appendices and subtle muscle movement to allow differentiated vocalization. Naked mole-rats are capable of many different vocalizations, but in general, lack fine motor control, although they do exhibit precise orientation movements in response to tactile stimuli of the vibrissae scattered all over their bodies [14,35]. This latter feature can be partially explained by the habitat within the confined body-sized burrows in which they reside and their greater reliance on teeth to expand burrow systems, as well as carry foods back to the nests.

Intelligent animals typically also exhibit complex social behavior, exemplified by primates and cetaceans, but also by eusocial organisms. This characteristic is so important that we are exploring it and its importance for intelligence in the next section in detail.

## 3. Eusociality, Intelligence, and the Naked Mole-Rat

Eusocial insects can accomplish impressive feats and the way they accomplish these is associated with their eusocial lifestyle. Examples include the elaborate structures built by termites which transform the environmental conditions they are exposed to by building structures with passive airflow equivalent to air conditioning. These energetically efficient and cooled large termite mounds are as large in proportion to the termite´s size as skyscrapers are to humans. The eusocial structure of the termite colony allows for such a complex feat of engineering, way beyond what could be accomplished by one termite on its own. However, Schulze-Makuch and Bains [19] suggested that the construction is not the result of information processing and then adapting actions to the local environment, but its instinctive, genetically determined, and a pre-programmed response similar to the eye-blink reflex in humans. Thus, we would expect that only mutations in their genetic code will enable them to adapt to new environmental conditions and building a different type of mound, not processes in the brain that we would associate with intelligent behavior. However, more recent research demonstrated an amazing behavioral and cognitive flexibility in bees and termites [36,37,38], and thus, it is controversial how much of the behavioral repertoire of a social insect is genetically “hard-wired” and how much of it can be attributed to observational learning capabilities similar to what we see in mammals. In either case, the feats that can be achieved with eusocial behavior are impressive, and one may wonder what other potential this evolutionary strategy has, especially if combined with a bigger brain of an individual organism. Similarly, naked mole-rats build a complex maze of burrows with both a variety of chambers and burrows at different depths. These enable mole-rats to select thermal gradients within which to bask and chambers within which to rest. This manipulation of the environment is a step along the path to intelligence, but not of the same magnitude as the intelligence in a chimpanzee or octopus.

Honeybees use a bee dance with each other to communicate precise information about direction and distance, accurately pointing out the direction of the food source, but also where it is located. Nevertheless, they cannot communicate any additional information, which might be useful, for example, are there other insects eating the food or is there a threat on the way to the food source, such as their common predator, the bee-eater bird. Therefore, the behavior of an individual bee shows little evidence of the plasticity we usually associate with intelligent behavior.

Ants engage in farming activities by cultivating fungi and keeping termites in captivity—not unlike how humans keep chicken or cattle. Similarly, naked mole-rats sustainably farm their largest food resource—*Pyrenacantha kaurbassana*. After boring into and consuming the nutrient-rich flesh, they block the holes they have created within the plant with soil, thereby not destroying these plant structures. They then allow these plant regions to regenerate, so they can continue feeding on them in situ.

Naked mole-rats are also eusocial, a phenomenon rarely seen in mammals, and have larger brains than bees or termites. Sociality cannot be underestimated as a pathway to intelligence. Our own species is the best example. Edward O. Wilson famously argued once that we as human beings are really “eusocial” apes, with the “eu” not representing a strict division of shared labors but meaning truly social, extremely cooperative and willing to make personal sacrifices for the greater good of the tribe [39]. Certainly, humans are social, not eusocial, but this provocative statement by Edward O. Wilson shows that we stand apart from other apes, and even from other hominids, with our brand of extreme “togetherness”. The social stability with accompanying group cohesion was favorable for the investment in complex behavior in humans [40], and this would also be expected in the eusocial naked mole-rats. Sociality is often demonstrated how the species takes care of their young and, as outlined above, care of young is also central to naked mole-rat eusociality.

A social behavior, which is closely related to intelligence, is play. The young of many intelligent animals play, and generally, the smarter they are, the more they play. Again, humans are an extreme example, because most of our activities are not obviously related to survival. Spinka et al. [41] emphasized that play enables animals to develop flexible kinematic and emotional responses to unexpected events. Fagan [42] understood play as improvised performance, with variations, of skilled motor and communicative actions in a context separate from the environment, meaning that play is doing things you would normally do, but not when and how you would normally do them [19]. Naked mole-rats are also very playful and engage in numerous play-like activities [43]. In determining their social hierarchy, juvenile naked mole-rats, in particular, participate in “play-fighting” in which they utilize their incisors, wrestle, have tug-of-wars often pushing each other over or may drag their siblings by the tail through the burrow system. This may involve pairs playing or even larger groups of individuals pushing, shoving and taking turns in these activities [44].

Another potential sign of intelligence is the use of tools to assist in energetically costly tasks. This ability to learn how to make and use tools is most obvious in primates [45], with chimpanzees reportedly capable of making specialized tools for digging out termites and ants, and even reportedly making spears for hunting [46]. Consistent use of tools has been rarely reported in rodents. However, both the naked mole-rat and its distant relative the degu reportedly have been shown to use tools [47,48]. Captive naked mole-rats may place tuber husks or wood shavings behind their incisors while gnawing on substrates and this is thought to prevent aspiration of foreign materials [47]. They also allegedly may occasionally pick up rocks or sticks to help with burrow excavation.

Hygienic behavior is another feature of intelligent organisms. Unlike most rodents, naked mole-rats defecate and urinate in a distinct chamber or toilet. In the wild, toilets are blind ending chambers relatively close to nest chambers and also scattered throughout the extensive maze of burrows. Latrines, when full, are plugged with soil and sealed and new toilets excavated nearby, in much the same way that “long drops” are used by humans. Similarly, in captivity naked mole-rats demarcate one or more (in larger colonies) chambers as toilets and undertake all their ablutions in this chamber. This hygienic behavior prevents their foods and bedding from being contaminated by fecal matter and the myriad of bacteria that may be present therein. They even transfer dead animals to the toilet chambers and may bury these animals in the dirty bedding material [44]. In these aspects, their level of hygiene is superior to that of certain human civilizations.

Another common feature shared by intelligent animals and closely linked to sociality and eusociality is the ability to communicate well with conspecifics. Humans spent an enormous amount of time on communication, and as a consequence, developed language. Naked mole-rats have at least 18 distinct vocalizations within our audible range—chirps, grunts, squeals, and hisses that communicate danger, anger, food, and the desire to mate [27,49]. This semblance of a language enables them to communicate their social hierarchy and may alert others within the group of chores that need to be done for the benefit of the colony. This extensive repertoire of vocalizations coupled with a plethora of other complex social behaviors suggest that while the brain of the naked mole-rat is relatively small for its size, it may pack a lot of punch as far as behavioral traits linked to intelligence are concerned.

## 4. Discussion and Conclusions

Intelligence is an extremely intangible phenomenon to objectively measure, linked to how well an animal responds to novel situations and constrained in no small part by the environmental constraints of its habitat. We have pointed out that compared to the laboratory rat and other mammals regarded as of higher intelligence, the EQ of naked mole-rats is rather low. However, we also pointed out that EQ is not a good comparable measure of intelligence between different species, and the brain of naked mole-rats has been rearranged and organized to be better equipped for the sensory modalities that can be employed in a dark and tightly constrained subterranean habitat. With both vision and long-distance hearing muffled underground, there is greater reliance on somatosensory perception. Not surprisingly, while the region of the brain dedicated to vision has markedly atrophied, the somatosensory cortex has expanded, albeit not to the same extent [33]. This overhaul of brain organization may contribute to their lower than expected EQ.

Another feature commonly associated with intelligence is dexterity, and this is associated with cerebellar volume. The cerebellum of the naked mole-rat appears to be in proportion to its brain size and naked mole-rats appear to lack dexterity. Unlike rats and mice that differentially use their fore and hind limbs, naked mole-rats tend to walk on all four legs, cannot jump and seldom use their paws for complex behaviors, although they frequently use their front paws to hold and manipulate foods while eating. While these features are likely constrained by the small radius of the burrows, little bigger than the size of the animals excavating these tunnels, based upon these features, we would not rate naked mole-rats as particularly intelligent animals. However, they have a few features that make us think that there is quite some potential for further development toward intelligence, perhaps even advanced intelligence.

While the EQ of naked mole-rats is rather low, several features suggest they may nevertheless be quite intelligent: as outlined above, they communicate widely among conspecifics, they are playful and anticipatory, with even some reports of tool use. The EQ appears to be similar in both breeders and non-breeders, but it was observed that the brain increases considerably in size in the queen and shows marked remodelling as the female becomes larger and heavier. Furthermore, neurogenesis is maintained well into adulthood [50,51,52], with evidence of extensive neurogenesis in animals entering their third decade [53], which is remarkable itself and shows that the brains of mature animals still retain enormous plasticity and are thus able to respond to changes in status and environment. If the enormous longevity of naked mole-rats is taken into account, then there may exist a clear pathway toward increasing intelligence.

The dominant breeding queen of a colony of naked mole-rats does not enter a menopause, rather the queens breed until they die. Moreover, they don´t seem to undergo senescence; and remain active throughout their long lives, ensuring the maintenance of dominance and breeding status. Queens may reign over their colony for several decades. Moreover, few offspring leave their natal colony—remaining in close proximity to the queen and pasha and serving the needs of their colony their entire lives. Therefore, in principle, they are available for much longer than other rodent parents to guide and train their offspring. Even breeding behavior and good parenting appears to be learnt, such that the offspring from stable colonies become better parents if given the opportunity to breed than the offspring from colonies where parenting is poor and *coup de etats* commonly happen. This prolonged influence over multiple litters of offspring may have contributed to their extreme longevity. A similar hypothesis has been proposed for other intelligent organisms (e.g., primates, cetaceans), which are very long-lived species for their body size: the grandmother hypothesis [54], here, grandmothers not only assist in the care of the young but provide key training that will assist the offspring in their own survival and thus benefit the entire social group [55]. This would be a very positive evolutionary trait if a potential invention needs to be communicated from one generation to another and beneficial behaviors (e.g., hygienic activities) and knowledge are learnt through intergenerational transfer.

An intriguing similarity between naked mole-rats and humans is that the colony size of naked mole-rats is about the same group size humans typically operate within. Dunbar [56] suggested that the number of neocortical neurons would limit the information-processing capacity of an organism and that this then would limit the number of relationships an individual can uphold simultaneously. As result, human groups, from Neolithic villages to today´s social networks, seem to work best when composed of no more than 100–150 individuals. A comparison of naked mole-rats with mice and humans in regard to the latent potential toward increased intelligence is shown in Table 1.

Of course, not all characteristics of naked mole-rats are encouraging for the potential future development of increased intelligence. A drawback is that their dexterity is quite poor. However, they are still able to construct kilometer-long tunnels, while, for example, the rather intelligent orca can only make waves. A huge advance for early humans to build civilizations was the domestication of animals. There is neither real potential nor evidence of such a development in naked mole-rats. They feed on enlarged structures that some plant species have in order to store nutrients [10]. However, they do not destroy these structures but allow the plant to regenerate, so they can continue feeding on them, which is essentially, a form of early farming and this is done sustainably—something which cannot be said about human farming (at least on a global scale). Another major advance for early humans was the control of an energy source: fire. While there is no such control of fire by naked mole-rats, fire is not a huge threat for them as they can hide in their burrow and can survive in very low oxygen conditions. Another marked achievement by humans was the development of language. Naked mole-rats do not have a language, but they have a high differentiation of sounds, probably very similar to the early humans when they developed language—associating certain sounds with different feelings and environmental scenarios.

One major difference between humans and naked mole-rats, pertaining to their potential of developing increased intelligence, is their habitat. While humans do take shelter in caves, and later in erected dwellings, naked mole-rats live underground, which also explains their small size and their poor vision. Life in the subsurface is restricted, and species are best adapted to their environment. Thus, having strong incisors is for them in their habitat certainly more important than an increase in intelligence.

However, even if we accept that there is a latent potential of naked mole-rats to develop a higher degree of intelligence, it does not mean that they will do so. As long as a stable environment is maintained, stabilizing selection will promote optimization of the current biological traits and elimination of peripheral values in the original population (Figure 2a). Under this scenario, an increase of intelligence would only be a peripheral value and any individual with an increased intelligence would not be favored in the natural selection process. The situation would become different with a changing environment, where natural selection would favor change in the direction that better adapts the organism to the new environment (Figure 2b). Certainly, many scenarios are imaginable and, for many of those, an increase in intelligence would not be a selective advantage, but there is one scenario which is similar to a scenario that played out on our planet about 65 million years ago and we will use that scenario as an instructive example.

Between 66 and 65 million years ago, a large asteroid impacted Earth resulting in the Chicxulub impact crater, which was, at least partially, responsible for the rise of the mammals after the demise of the dinosaurs. Although it is now recognized that the diversity of mammals in the Cretaceous Period was much larger than previously realized, many of those mammals were nocturnal rodent-like species ranging in size from mice to juvenile rats. Due to newly available habitats on the surface of our planet that opened up after the extinction of the non-avian dinosaurs, these mammals further diversified to the variety of shapes and sizes seen today, with great evolutionary success. If such an event or a similar event (for example, triggered by a nearby supernova explosion) would re-occur, with the result of rendering the planetary surface poorly habitable for an extended period of time, naked mole-rats would have some abilities, including their longevity, plasticity, sustainable farming ability, social cohesion and interaction, perhaps even their eusociality, that might allow them to take advantage of this scenario (Table 1). Not all behavioral patterns would be advantageous though. One obvious challenge for the naked mole-rat would be that it is so dependent on the plant species it feeds off. Plant species like *Pyrenacantha kaurbassana* would be starkly affected by such a calamity, perhaps even become extinct, and the naked mole-rat would have to adapt to alternative food sources or become extinct. Eusocial species, however, have been shown to respond quickly to natural selection pressures by rapidly evolving genes [58], and if the natural selection pressure would force the species to live closer to the surface and later at the surface, in order to recover new food sources, the more heterogeneous environment would likely promote adaptive abilities, including increased intelligence, during and after the transition.

How far the naked mole-rat may or a species similar to it may progress toward truly advanced intelligence is difficult to say and goes beyond the scope of this paper. However, if it would adapt to feed on other food sources (e.g., underground fungi or scavenging), the naked mole-rat might also serve as a macroscopic and multicellular model organism for a planet or moon on which the surface became permanently uninhabitable (after the evolution of animals already occurred).

## Figures and Tables

**Figure 1 life-09-00076-f001:**
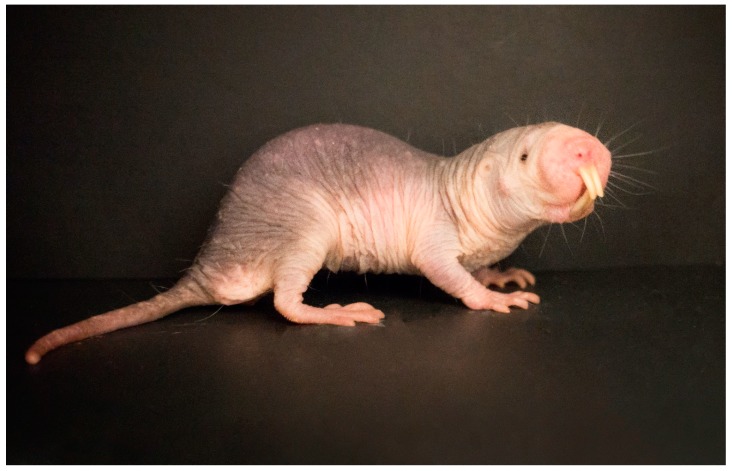
The naked mole-rat shown in the photograph is 10 cm long and weighs 40 g. (Picture taken by Rochelle Buffenstein, Calico Life Sciences LLC).

**Figure 2 life-09-00076-f002:**
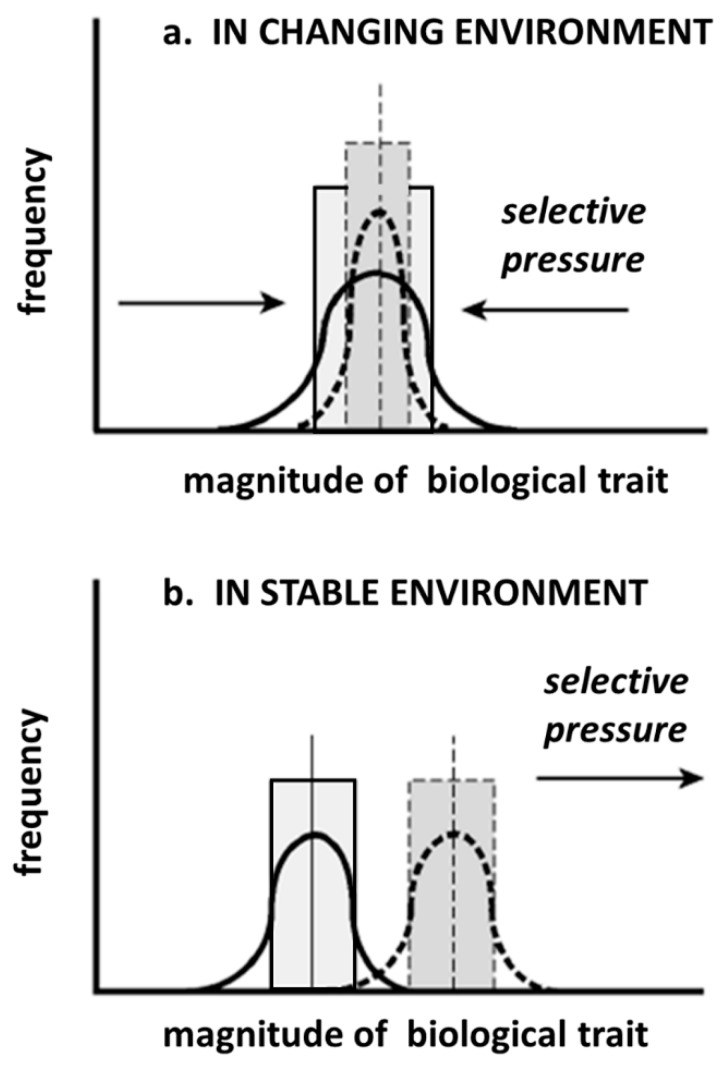
Effect of selective pressure on biological characteristics, illustrated by changes in the frequency distribution of a quantitative biological trait in response to different forms of selective pressure. (**a**) In changing environments, natural selection favors change in the direction that better adapts the organism to the new environment. The range for the majority of organisms from the original population (solid rectangle) and their mean value (solid vertical line) shift toward a different mean (dashed vertical line) without changing the range of the variable in the new population (dashed rectangle). The biological trait could also be intelligence (**b)** In stable environments, stabilizing selection promotes elimination of peripheral values in the original population, reducing the range in the descendent population (dashed rectangle) without altering the mean value (dashed vertical line). (adapted from Schulze-Makuch and Irwin [57]).

**Table 1 life-09-00076-t001:** Comparison of factor thought to be important for the development of intelligence.

Organism	Longevity	Social	Large Brain	Farming/Sustainability	Sanitary	Avoid Senescence	Language
Naked Mole-Rat	✔	✔	✘ ^1^	✔	✔^.^^3^	✔	✘ ^4^
Mouse	✘	✘	✘	✘	✘	✘	✘
Homo sapiens	✔	✔	✔	✔^.^^2^	✔	✘	✔

Notes: ^1^ naked mole-rats have a small brain. However, that of the queen is larger and can create new neurons during her lifetime, if needed; ^2^ it is debatable whether humans do actually perform sustainable farming. They are capable to do so in principle, but considering the global population, this is not the case; ^3^ on the individual and the population level, naked mole-rats keep more sanitary conditions than humans; ^4^ not what we would call language, but a rich repertoire of sounds for different situations.

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
