# Peer review of "The Naked Mole-Rat: An Unusual Organism with an Unexpected Latent Potential for Increased Intelligence?"

_life, 2019, doi:10.3390/life9030076_

Round 1
Reviewer 1 Report
I found this to be a very interesting essay. I only have minor comments/suggestions.
1) Abstract. "Although their encephalization quotient, a marker of intelligence..." Perhaps consider changing to, "Although their encephalization quotient, a controversial marker of intelligence..."
2) Line 30. Should damaraland be Damaraland?
3) Line 45. "Park et al., 2017" is in the text. Should it be included in the References?
4) Figure 1. Should there be a photo credit?
5) Line 141. "mole-rats" should be "naked mole-rats"
6) Line 148. Re motor control: Would the author consider reasonably precise orientation movements toward tactile stimuli to be an example of fine motor control (see Crish, et al., 2003 Brain Behav Evol. 2003;62(3):141-51). If not, please ignore this question.
7) Line 219. "allegedly may occasionally pick up rocks or sticks to help with burrow excavation..." Is there a reference (either published or unpublished?
8) Line 228. "They even transfer dead animals to the toilet chamber..." I have also heard that rumor but I have not been able to find a reference. If you have a reference, please include it. Otherwise you might want to soften the sentence to something like, "There are even rumors that they occasionally transfer dead animals to the toilet chamber..."
9) Line 238. "the brain of the naked mole-rat is small for its size, it may pack a lot of punch..." I suggest changing to, "the brain of the naked mole-rat is relatively small for its size, it may pack a lot of punch..."
10) Line 255. "seldom use their paws for complex behaviors." I have frequently observed them to lie down and use their front paws to hold and manipulate food items while eating. I don't know if this would be considered a complex behavior. Author's discretion.
11) Line 265. "Furthermore, neurogenesis is maintained throughout their whole life time..." This should have a reference. Perhaps: Orr ME, Garbarino VR, Salinas A, Buffenstein R. (2016) Extended Postnatal Brain Development in the Longest-Lived Rodent: Prolonged Maintenance of Neotenous Traits in the Naked Mole-Rat Brain. Front Neurosci. 2016 Nov 8;10:504. eCollection 2016. In which case, "throughout their whole life time" is inaccurate. Better to say, "into adulthood" since they only tested up to 3 years. But maybe the author has a reference that I'm not aware of?
Reviewer 2 Report
The manuscript by Schulze-Makuch submitted to “Life” focuses on the different aspects that can be considered to define the naked mole rat (NMR) an intelligent animal, such as active life-style, distinct vocalizations, manipulation of environment, usage of tools for hard tasks and hygienic behavior.
This review article is well written and relevant since it expands the knowledge about the naked mole rat deeply describing its peculiar characteristics and widely explaining the different factors which can led to consider it an intelligent animal. This publication will also be useful for researchers working on this species. Of interest, the hypothesis about a possible survival of this animal species in extreme conditions could work as a starting point for new analyses about the great potential of this animal (as a model) in harsh situations.
On the whole there is maybe a major pitfall: many features of NMR the Authors deal with, particularly “intelligence”, are not correctly posed in an evolutionary context. Most of these features are simply long-lasting adaptations to their particular environment and thus it is not utterly correct (maybe useless) to make comparisons with other distant groups (e.g., humans and dolphins). Can the Authors try to refine such concepts?
Here following, some suggestions to the Authors about some minor points needing a revision:
Line 24: It’s true that naked mole rat is well known for its resistance to cancer, but it could be interesting to cite two papers showing presence of cancer in the naked mole rat (Taylor et al., 2016, J. Gerontol. A Biol. Sci. Med. Sci.; Delaney et al., 2016, Vet. Pathol.)
Lines 66 to 68: Add references about the link between intelligence and how much widely an animal need to search for food
Lines 106 to 108: It’s true that generally large-sized animals show intelligence, but also small animals can show it (like the red fox) so adding a reference here could be useful
Lines 221 to 230: Also add a reference about the link between hygienic behavior and intelligence. Is the naked mole rat behavior that is mentioned in the article either something that you see in your colonies or something found in literature?
Lines 263 to 266: Add a reference about neurogenesis (better: adult neurogenesis) in naked mole rat; some reports deeply analyzed the neurogenic process in this species (Penz et al., 2015, Sci. Rep.), as well as a reference about the brain size of the queen.
